# Doubly Nonnegative and Semidefinite Relaxations for the Densest *k*-Subgraph Problem

**DOI:** 10.3390/e21020108

**Published:** 2019-01-24

**Authors:** Chuan-Hao Guo, Yuan Guo, Bei-Bei Liu

**Affiliations:** School of Economics and Management, Zhejiang Sci-Tech University, Hangzhou 310018, China

**Keywords:** densest *k*-subgraph, doubly nonnegative relaxation, semidefinite relaxation, approximation ratio, NP-hard, 90C10, 90C20, 49M20, 65K05

## Abstract

The densest *k*-subgraph (DkS) maximization problem is to find a set of *k* vertices with maximum total weight of edges in the subgraph induced by this set. This problem is in general NP-hard. In this paper, two relaxation methods for solving the DkS problem are presented. One is doubly nonnegative relaxation, and the other is semidefinite relaxation with tighter relaxation compare with the relaxation of standard semidefinite. The two relaxation problems are equivalent under the suitable conditions. Moreover, the corresponding approximation ratios’ results are given for these relaxation problems. Finally, some numerical examples are tested to show the comparison of these relaxation problems, and the numerical results show that the doubly nonnegative relaxation is more promising than the semidefinite relaxation for solving some DkS problems.

## 1. Introduction

In this paper, the densest *k*-subgraph (DkS) problem [1,2] is considered. For a given graph *G* and a parameter *k*, the DkS problem consists in finding a maximal average degree in the subgraph induced by the set of *k* vertices. This problem was first introduced by Corneil and Perl as a natural generalization of the maximum clique problem [3]. It is NP-hard on restricted graph classes such as chordal graphs [3], bipartite graphs [3] and planar graphs [4]. The DkS problem is a classical problem of combinatorial optimization and arises in several applications, such as facility location [5], community detection in social networks, identifying protein families and molecular complexes in protein–protein interaction networks [6], etc. Since the DkS problem is in general NP-hard, there are a few approximation methods [7,8,9] for solving it. It is well-known that semidefinite relaxation is a powerful and computationally efficient approximation technique for solving a host of very difficult optimization problems, for instance, the max-cut problem [10] and the boolean quadratic programming problem [11]. It also has been at the center of some of the very exciting developments in the area of signal processing [12,13].

Optimization problems over the doubly nonnegative cone arise, for example, as a strengthening of the Lovasz-ϑ-number for approximating the largest clique in a graph [14]. The recent work by Burer [15] stimulated the interest in optimization problems over the completely positive cone. A tractable approximation to such problem being defined as an optimization problem over the doubly nonnegative cone. By using the technique of doubly nonnegative relaxation, Bai and Guo proposed an effective and promising method for solving multiple objective quadratic programming problems in [16]. For more details and developments of this technique, one may refer to [17,18,19] and the references therein. It is worth pointing out that the cone of doubly nonnegative matrices is a subset of a positive semidefinite matrices cone. Thus, the doubly nonnegative relaxation is more promising than the basic semidefinite relaxation. Moreover, such relaxation problems can be efficiently solved by some popular package software.

In this paper, motivated by the idea of doubly nonnegative relaxation and semidefinite relaxation, the two relaxation methods for solving the DkS problem are presented. One is doubly nonnegative relaxation, and the other is semidefinite relaxation with tighter relaxation. Furthermore, we prove that the two relaxation problems are equivalent under the suitable conditions. Some approximation accuracy results about these relaxation problems are also given. Finally, we report some numerical examples to show the comparison of the two relaxation problems. The numerical results show that the doubly nonnegative relaxation is more promising than the semidefinite relaxation for solving some DkS problems.

The paper is organized as follows: we present doubly nonnegative relaxation and a new semidefinite relaxation with tighter relaxation for the DkS problem in Section 2.1 and Section 2.2, respectively. In Section 3, we prove that the two new relaxations proposed in Section 2 are equivalent. In Section 4, some approximation accuracy results for the proposed relaxation problems are given. Some comparative numerical results are reported in Section 5 to show the efficiency of the proposed new relaxations. Moreover, some concluding remarks are given in Section 6.

## 2. Two Relaxations for the Densest k-Subgraph Problem

First of all, the definition of the densest *k*-subgraph (DkS) problem is given as follows.

**Definition** **1** (Densest *k*-subgraph)**.**
*For a given graph G(V,E), where V is the vertex set and E is the edge set. The DkS problem on G(V,E) is the problem of finding a vertex subset of V of size k with the maximum induced average degree.*


Given a symmetric n×n matrix $A=(a_{ij})$, the weighted graph with vertex set {1,2,…,n} associates with *A* in such a way: the edge [i,j] with the weight $a_{ij}$ is introduced in the graph. Then, *A* is interpreted as the weighted adjacency matrix of the graph with the vertex set V={1,2,…,n}. Based on Definition 1, the DkS problem consists of determining a subset $V_{1}\subseteq V$ consisting of *k* vertices such that the total weight of edges in the subgraph spanned by $V_{1}$ is maximized. To select subgraphs, assign a decision variable $y_{i}\in \left\{0,1\right\}$ for each node ($y_{i}=1$ if the node is taken, and $y_{i}=0$ if the node is not). The weight of the subgraph given by *y* is $y^{T}Ay$. Thus, the DkS problem can be phrased as the 0−1 quadratic problem
$$\left(\mathrm{DkS}\right)\;\;\;\;\begin{array}{cc} \max & \frac{1}{2}y^{T}Ay,\\ s.t. & y^{T}e=k,\\ & y\in {\left\{0,1\right\}}^{n}. \end{array}$$

It is known that the (DkS) problem is NP-hard [20], even though *A* is assumed to be positive semidefinite, since the feasible space of the (DkS) problem is nonconvex. For solving this problem efficiently, we present the two new relaxations for the (DkS) problem in the following subsections, based on the idea of approximation methods.

### 2.1. Doubly Nonnegative Relaxation

Note that the quadratic term $y^{T}Ay$ in the (DkS) problem can also be expressed as $A•yy^{T}$. By introducing a new variable $Y=yy^{T}$ and taking lifting techniques, we could reformulate the (DkS) problem into the following completely positive programming problem:$$\left({\mathrm{CPP}}_{\mathrm{DkS}}\right)\;\;\;\;\begin{array}{cc} \max & \frac{1}{2}A•Y,\\ s.t. & y^{T}e=k,\\ & e^{T}Ye=k^{2},\\ & \mathrm{diag}(Y)=y,\\ & \left[\begin{array}{cc} 1 & y^{T}\\ y & Y \end{array}\right]\in C_{1+n}, \end{array}$$where $C_{1+n}$ is defined as follows:$$C_{1+n}:=\left\{X\in S_{1+n}:\;X=\sum_{h\in H}z^{h}{\left(z^{h}\right)}^{T}\right\}\cup \left\{0\right\},$$and for some finite vectors ${\left\{z^{h}\right\}}_{h\in H}\subset R_{1+n}^{+}\backslash \left\{0\right\}$.

The following theorem shows the relationship between the (DkS) problem and the $\left({\mathrm{CPP}}_{\mathrm{DkS}}\right)$ problem. Its proof is similar to the one of Theorem 2.6 in [15] and is omitted here.

**Theorem** **1.**
*(i) The (DkS) problem and the $\left({\mathrm{CPP}}_{\mathrm{DkS}}\right)$ problem have the same optimal values of objective functions, i.e., $\mathrm{Opt}(\mathrm{DkS})=\mathrm{Opt}({\mathrm{CPP}}_{\mathrm{DkS}})$; (ii) if $\left(y^{*},Y^{*}\right)$ is an optimal solution for the $\left({\mathrm{CPP}}_{\mathrm{DkS}}\right)$ problem; then, $y^{*}$ is in the convex hull of optimal solutions for the (DkS) problem.*


On one hand, according to Definition 2, it is obviously that the $\left({\mathrm{CPP}}_{\mathrm{DkS}}\right)$ problem is equivalent to the (DkS) problem. On the other hand, in view of the definition of convex cone, $C_{1+n}$ is a closed convex cone, and is called completely positive matrices cone. Thus, the $\left({\mathrm{CPP}}_{\mathrm{DkS}}\right)$ problem is convex. However, since checking whether or not a given matrix belongs to $C_{1+n}$ is NP-hard, which has been shown by Dickinson and Gijen in [21], the $\left({\mathrm{CPP}}_{\mathrm{DkS}}\right)$ problem is still NP-hard. Thus, $C_{1+n}$ has to be replaced or approximated by some computable cones. For example, $R_{n}^{+}$ and $S_{n}^{+}$ are both computable cones; furthermore, $N_{n}^{+}$ is also a computable cone.

It is worth mentioning that Diananda’s decomposition theorem [22] can be reformulated as follows, and its proof can be found in it.

**Theorem** **2.**
*$C_{n}\subseteq S_{n}^{+}\cap N_{n}^{+}$ holds for all n. If n≤4, then $C_{n}=S_{n}^{+}\cap N_{n}^{+}$.*


The matrices cone $S_{n}^{+}\cap N_{n}^{+}$ is sometimes called “doubly nonnegative matrices cone”. Of course, in dimension n≥5, there are matrices which are doubly nonnegative but not completely positive, the counterexample can be seen in [23].

By using Theorem 2, the $\left({\mathrm{CPP}}_{\mathrm{DkS}}\right)$ problem can be relaxed to the problem
$$\left({\mathrm{DNNP}}_{\mathrm{DkS}}\right)\;\;\;\;\begin{array}{cc} \max & \frac{1}{2}A•Y,\\ s.t. & y^{T}e=k,\\ & e^{T}Ye=k^{2},\\ & \mathrm{diag}(Y)=y,\\ & \left[\begin{array}{cc} 1 & y^{T}\\ y & Y \end{array}\right]\in S_{1+n}^{+}\cap N_{1+n}^{+}, \end{array}$$which is called the doubly nonnegative relaxation for the (DkS) problem. Some explanations are given below for this relaxation problem.

**Remark** **1.**
*Obviously, the $\left({\mathrm{DNNP}}_{\mathrm{DkS}}\right)$ problem has a linear objective function and the linear constraints as well as a convex conic constraint, so it is a linear conic programming problem. Meanwhile, it is notable that $S_{1+n}^{+}\cap N_{1+n}^{+}\subseteq S_{1+n}^{+}$ and the types of variables in both the sets are the same, which further implies that the $\left({\mathrm{DNNP}}_{\mathrm{DkS}}\right)$ problem could be solved by some popular package softwares for solving semidefinite programs.*


### 2.2. New Semidefinite Relaxation

It is well-known that semidefinite relaxation is a powerful approximation technique for solving a host of combinatorial optimization problems. In this subsection, we present a new semidefinite relaxation with tighter bound for the (DkS) problem.

The idea of the standard lifting is to introduce the symmetric matrix of rank one $Y=yy^{T}$. With the help of *Y*, we could express the integer constraints $y_{i}\in \left\{0,1\right\}$ as $Y_{ii}=y_{i}$, and the quadratic objective function $y^{T}Ay$ as A•Y. Thus, we can get the following equivalent formulation of the (DkS) problem
$$\begin{array}{cc} \max & \frac{1}{2}A•Y,\\ s.t. & y^{T}e=k,\\ & e^{T}Ye=k^{2},\\ & Y=yy^{T},\\ & \mathrm{diag}(Y)=y,\\ & \mathrm{rank}(Y)=1. \end{array}$$

Notice then that the hard constraint in the above problem is the constraint rank(Y)=1, which is moreover difficult to handle. Thus, we can relax the above problem to the following standard semidefinite relaxation problem by dropping the rank-one constraint
$$\left(I-{\mathrm{SDR}}_{\mathrm{DkS}}\right)\;\;\;\;\begin{array}{cc} \max & \frac{1}{2}A•Y,\\ s.t. & y^{T}e=k,\\ & e^{T}Ye=k^{2},\\ & \mathrm{diag}(Y)=y,\\ & \left[\begin{array}{cc} 1 & y^{T}\\ y & Y \end{array}\right]\in S_{1+n}^{+}. \end{array}$$

For the $\left(I-{\mathrm{SDR}}_{\mathrm{DkS}}\right)$ problem, some remarks are given below.

**Remark** **2.**
*(i) Obviously, the $\left(I-{\mathrm{SDR}}_{\mathrm{DkS}}\right)$ problem is also a linear conic programming problem, it has the same objective function and the equality constraints with the $\left({\mathrm{DNNP}}_{\mathrm{DkS}}\right)$ problem. The only difference between the $\left(I-{\mathrm{SDR}}_{\mathrm{DkS}}\right)$ problem and the $\left({\mathrm{DNNP}}_{\mathrm{DkS}}\right)$ problem is that the $\left({\mathrm{DNNP}}_{\mathrm{DkS}}\right)$ problem has $\frac{n(n+1)}{2}+n$ nonnegative constraints more than the $\left(I-{\mathrm{SDR}}_{\mathrm{DkS}}\right)$ problem.*

*(ii) Since $S_{1+n}^{+}\cap N_{1+n}^{+}\subseteq S_{1+n}^{+}$, it holds that $\mathrm{Opt}\left({\mathrm{DNNP}}_{\mathrm{DkS}}\right)\leq \mathrm{Opt}\left(I-{\mathrm{SDR}}_{\mathrm{DkS}}\right)$ from Remark 2 (i). Thus, the bound of the $\left({\mathrm{DNNP}}_{\mathrm{DkS}}\right)$ problem is not larger than the one of the $\left(I-{\mathrm{SDR}}_{\mathrm{DkS}}\right)$ problem. In Section 5, we implement some numerical experiments to show the comparison between the $\left(I-{\mathrm{SDR}}_{\mathrm{DkS}}\right)$ problem and the $\left({\mathrm{DNNP}}_{\mathrm{DkS}}\right)$ problem from the computational point of view.*


Note that the (DkS) problem is inhomogeneous, but we can homogenize it as follows. First, let z=2y−e in the (DkS) problem, it follows that $z\in {\left\{-1,1\right\}}^{n}$. Thus, the change of variable y→z gives the following equivalent formulation of the (DkS) problem:$$\hat{\left(\mathrm{DkS}\right)}\;\;\;\;\begin{array}{cc} \max & \frac{1}{8}z^{T}Az+\frac{1}{4}z^{T}Ae+\frac{1}{8}e^{T}Ae,\\ s.t. & z^{T}e=2k-n,\\ & z\in {\left\{-1,1\right\}}^{n}. \end{array}$$

Then, with the introduction of the extra variable *t*, the (DkS) problem can be expressed as a homogeneous problem
$$\bar{\left(\mathrm{DkS}\right)}\;\;\;\;\begin{array}{cc} \max & \left[\begin{array}{cc} t & z^{T} \end{array}\right]\left[\begin{array}{cc} \frac{1}{8}e^{T}Ae & \frac{1}{8}e^{T}A\\ \frac{1}{8}Ae & \frac{1}{8}A \end{array}\right]\left[\begin{array}{c} t\\ z \end{array}\right],\\ s.t. & \left[\begin{array}{cc} t & z^{T} \end{array}\right]\left[\begin{array}{cc} 0 & \frac{1}{2}e^{T}\\ \frac{1}{2}e & 0 \end{array}\right]\left[\begin{array}{c} t\\ z \end{array}\right]=2k-n,\\ & \left[\begin{array}{c} t\\ z \end{array}\right]\in {\left\{-1,1\right\}}^{1+n}, \end{array}$$where 0 is a zero matrix with appropriate dimension.

**Remark** **3.**
*The $\bar{\left(\mathrm{DkS}\right)}$ problem is equivalent to the $\hat{\left(\mathrm{DkS}\right)}$ problem in the following sense: if $\left[\begin{array}{c} t^{*}\\ z^{*} \end{array}\right]$ is an optimal solution to the $\bar{\left(\mathrm{DkS}\right)}$ problem, then $z^{*}$ (resp. $-z^{*}$) is an optimal solution to the $\hat{\left(\mathrm{DkS}\right)}$ problem with $t^{*}=1$ (resp. $t^{*}=-1$).*


By using the standard semidefinite relaxation technique, and letting let $S=\left[\begin{array}{c} t\\ z \end{array}\right]\left[\begin{array}{cc} t & z^{T} \end{array}\right]$, the $\bar{\left(\mathrm{DkS}\right)}$ problem can be relaxed to the following problem:$$\left({\mathrm{SDR}}_{\bar{\mathrm{DkS}}}\right)\;\;\;\;\begin{array}{cc} \max & \left[\begin{array}{cc} \frac{1}{8}e^{T}Ae & \frac{1}{8}e^{T}A\\ \frac{1}{8}Ae & \frac{1}{8}A \end{array}\right]•S,\\ s.t. & \left[\begin{array}{cc} 0 & \frac{1}{2}e^{T}\\ \frac{1}{2}e & 0 \end{array}\right]•S=2k-n,\\ & \mathrm{diag}(S)=e,\\ & S\in S_{1+n}^{+}. \end{array}$$

Moreover, again by using the standard semidefinite relaxation technique directly to the $\hat{\left(\mathrm{DkS}\right)}$ problem, we have from $Z=zz^{T}$,
$$\left({\mathrm{SDR}}_{\hat{\mathrm{DkS}}}\right)\;\;\;\;\begin{array}{cc} \max & \frac{1}{8}A•Z+\frac{1}{4}z^{T}Ae+\frac{1}{8}e^{T}Ae,\\ s.t. & z^{T}e=2k-n,\\ & e^{T}Ze={\left(2k-n\right)}^{2},\\ & \mathrm{diag}(Z)=e,\\ & \left[\begin{array}{cc} 1 & z^{T}\\ z & Z \end{array}\right]\in S_{1+n}^{+}. \end{array}$$

The $\left({\mathrm{SDR}}_{\bar{\mathrm{DkS}}}\right)$ problem and the $\left({\mathrm{SDR}}_{\hat{\mathrm{DkS}}}\right)$ problem are both standard semidefinite relaxation problems for the (DkS) problem. The upshot of the formulations of these two relaxation problems is that they can be solved very conveniently and efficiently, to some arbitrary accuracy, by some readily available software packages, such as CVX. Note that there is only one difference between these two relaxation problems, i.e., the $\left({\mathrm{SDR}}_{\hat{\mathrm{DkS}}}\right)$ problem has one equality constraint more than the $\left({\mathrm{SDR}}_{\bar{\mathrm{DkS}}}\right)$ problem. In Section 5, some comparative numerical results are reported to show the effectiveness of these two relaxations problems for solving some random (DkS) problems, respectively.

It is worth noting that the constraint $z\in {\left\{-1,+1\right\}}^{n}$ in the $\hat{\left(\mathrm{DkS}\right)}$ problem further implies
(1)$$\left(1+z_{i}\right)\left(1+z_{j}\right)\geq 0\Rightarrow 1+z_{i}+z_{j}+Z_{ij}\geq 0,\;\forall 1\leq i\leq j\leq n$$always holds. Thus, adding Formula (Equation 1) to the (${\mathrm{SDR}}_{\hat{\mathrm{DkS}}}$) problem, we come up with the following new semidefinite relaxation problem
$$\left(\mathrm{II}-{\mathrm{SDR}}_{\mathrm{DkS}}\right)\;\;\;\;\begin{array}{cc} \max & \frac{1}{8}A•Z+\frac{1}{4}z^{T}Ae+\frac{1}{8}e^{T}Ae,\\ s.t. & z^{T}e=2k-n,\\ & e^{T}Ze={\left(2k-n\right)}^{2},\\ & \mathrm{diag}(Z)=e,\\ & 1+z_{i}+z_{j}+Z_{jj}\geq 0,\;\forall 1\leq i\leq j\leq n,\\ & \left[\begin{array}{cc} 1 & z^{T}\\ z & Z \end{array}\right]\in S_{1+n}^{+}. \end{array}$$

Obviously, the relationship $\mathrm{Opt}\left(\mathrm{II}-{\mathrm{SDR}}_{\mathrm{DkS}}\right)\leq \mathrm{Opt}\left({\mathrm{SDR}}_{\hat{\mathrm{DkS}}}\right)$ holds since the feasible set of the $\left(\mathrm{II}-{\mathrm{SDR}}_{\mathrm{DkS}}\right)$ problem is the subset of the feasible set of the $\left({\mathrm{SDR}}_{\hat{\mathrm{DkS}}}\right)$ problem and the two problems have the same objective function.

Up to now, three new semidefinite relaxation problems for the (DkS) problem are established, i.e., the $\left({\mathrm{SDR}}_{\bar{\mathrm{DkS}}}\right)$ problem, the (${\mathrm{SDR}}_{\hat{\mathrm{DkS}}}$) problem and the $\left(\mathrm{II}-{\mathrm{SDR}}_{\mathrm{DkS}}\right)$ problem, in which the upper bound of the $\left(\mathrm{II}-{\mathrm{SDR}}_{\mathrm{DkS}}\right)$ problem is more promising than the one of the (${\mathrm{SDR}}_{\hat{\mathrm{DkS}}}$) problem. In the following sections, we will further investigate the relationship between these three problems with the $\left({\mathrm{DNNP}}_{\mathrm{DkS}}\right)$ problem.

## 3. The Equivalence between the Relaxation Problems

The previous section establishes the doubly nonnegative relaxation (i.e., the $\left({\mathrm{DNNP}}_{\mathrm{DkS}}\right)$ problem) and the semidefinite relaxation with tighter bound (i.e., the $\left(\mathrm{II}-{\mathrm{SDR}}_{\mathrm{DkS}}\right)$ problem) for the (DkS) problem. Note that the $\left({\mathrm{DNNP}}_{\mathrm{DkS}}\right)$ problem has *n* inequality constraints more than the $\left(\mathrm{II}-{\mathrm{SDR}}_{\mathrm{DkS}}\right)$ problem. In this section, we will prove the equivalence between the two relaxations. First of all, the definition of the equivalence of two optimization problems is given as follows.

**Definition** **2.**
*Two problems P1 and P2 are called equivalent if they satisfy the following two conditions:*

*(i) Opt(P1)=Opt(P2).*

*(ii) If from a solution of P1, then a solution of P2 is readily found, and vice versa.*


In order to establish the equivalence for the $\left({\mathrm{DNNP}}_{\mathrm{DkS}}\right)$ problem and the $\left(\mathrm{II}-{\mathrm{SDR}}_{\mathrm{DkS}}\right)$ problem, a crucial theorem is given below and the details of its proof can be seen in [24] (Appendix A.5.5).

**Theorem** **3** (Schur Complement)**.**
*Let matrix $M\in S^{n}$ is partitioned as*
$$M=\left[\begin{array}{cc} D & B\\ B^{T} & C \end{array}\right].$$

*If detD≠0, the matrix $H=C-B^{T}D^{-1}B$ is called the Schur complement of D in M. Then, the following relations hold:*

*(i) M≻0 if and only if D≻0 and H≻0.*

*(ii) If D≻0, then M⪰0 if and only if H⪰0.*


To the end, by using Definition 2 and Theorem 3, we have the following main equivalence theorem.

**Theorem** **4.**
*Suppose that the feasible sets $\mathrm{Fea}({\mathrm{DNNP}}_{\mathrm{DkS}})$ and $\mathrm{Fea}\left(\mathrm{II}-{\mathrm{SDR}}_{\mathrm{DkS}}\right)$ are both nonempty. Then, the $\left({\mathrm{DNNP}}_{\mathrm{DkS}}\right)$ problem and the $\left(\mathrm{II}-{\mathrm{SDR}}_{\mathrm{DkS}}\right)$ problem are equivalent.*


**Proof.** First of all, we prove that $\mathrm{Opt}\left({\mathrm{DNNP}}_{\mathrm{DkS}}\right)\geq \mathrm{Opt}\left(\left(\mathrm{II}-{\mathrm{SDR}}_{\mathrm{DkS}}\right)\right)$.Suppose that $\left(z^{*},Z^{*}\right)$ is an optimal solution of the $\left(\mathrm{II}-{\mathrm{SDR}}_{\mathrm{DkS}}\right)$ problem, and let
(2)$$Y=\frac{1}{4}\left(ee^{T}+e{\left(z^{*}\right)}^{T}+z^{*}e^{T}+Z^{*}\right),\;y=\frac{1}{2}\left(e+z^{*}\right).$$Directly from $e^{T}z^{*}=2k-n$ and Equation (Equation 2), we have
(3)$$y^{T}e=\frac{1}{2}{\left(e+z^{*}\right)}^{T}e=\frac{1}{2}\left(n+e^{T}z^{*}\right)=k.$$By Equation (Equation 2) and $e^{T}Z^{*}e={\left(2k-n\right)}^{2}$, it holds that
(4)$$\begin{array}{cc} e^{T}Ye & =\frac{1}{4}e^{T}\left(ee^{T}+e{\left(z^{*}\right)}^{T}+\left(z^{*}\right)e^{T}+Z^{*}\right)e\\ & =\frac{1}{4}\left(n^{2}+2n{\left(z^{*}\right)}^{T}e+{\left(2k-n\right)}^{2}\right)\\ & =k^{2}. \end{array}$$Since $\mathrm{diag}(Z^{*})=e$, Equation (Equation 2) further implies that
(5)$$\mathrm{diag}\left(Y\right)=\frac{1}{4}(e+2z^{*}+\mathrm{diag}\left(Z^{*}\right)=\frac{1}{2}\left(e+z^{*}\right)=y.$$From $1+z_{i}^{*}+z_{j}^{*}+Z_{ij}^{*}\geq 0$ for all 1≤i≤j≤n, it holds that
(6)$$Y_{ij}\geq 0,\forall 1\leq i\leq j\leq n.$$Combining with Equation (Equation 5), it is true that from Formula (Equation 6)
(7)$$y_{i}\geq 0,\forall 1\leq i\leq n.$$By Theorem 3 (ii) and Equation (Equation 2), it follows that
$$\begin{array}{cc} Y-yy^{T} & =\frac{1}{4}\left(ee^{T}+e{\left(z^{*}\right)}^{T}+z^{*}e^{T}+Z^{*}\right)-\frac{1}{4}\left(e+z^{*}\right){\left(e+z^{*}\right)}^{T}\\ & =\frac{1}{4}\left(Z^{*}-z^{*}{\left(z^{*}\right)}^{T}\right)⪰0, \end{array}$$i.e.,
(8)$$\left[\begin{array}{cc} 1 & y^{T}\\ y & Y \end{array}\right]\in S_{1+n}^{+}.$$From Equations (Equation 3), (Equation 4) and (Equation 6)–(Equation 8), we have (y,Y) defined by Equation (Equation 2) is a feasible solution of the $\left({\mathrm{DNNP}}_{\mathrm{DkS}}\right)$ problem. Moreover, again from Equation (Equation 2), we obtain
$$\begin{array}{cc} \frac{1}{2}A•Y & =\frac{1}{8}A•\left(ee^{T}+e{\left(z^{*}\right)}^{T}+z^{*}e^{T}+Z^{*}\right)\\ & =\frac{1}{8}A•Z^{*}+\frac{1}{4}{\left(z^{*}\right)}^{T}Ae+\frac{1}{8}e^{T}Ae\\ & =\mathrm{Opt}(\left(\mathrm{II}-{\mathrm{SDR}}_{\mathrm{DkS}}\right)), \end{array}$$i.e., $\mathrm{Opt}\left({\mathrm{DNNP}}_{\mathrm{DkS}}\right)\geq \mathrm{Opt}\left(\left(\mathrm{II}-{\mathrm{SDR}}_{\mathrm{DkS}}\right)\right)$.Conversely, given an optimal solution $\left(y^{*},Y^{*}\right)$ to the $\left({\mathrm{DNNP}}_{\mathrm{DkS}}\right)$ problem, and let
(9)$$Z_{ij}=1-2y_{i}^{*}-2y_{j}^{*}+4Y_{ij}^{*},\;z_{i}=2y_{i}^{*}-1,\;\forall 1\leq i\leq j\leq n.$$Since $\mathrm{diag}\left(Y^{*}\right)=y^{*}$, Equation (Equation 9) further implies that
(10)$$\mathrm{diag}\left(Z\right)=e-4y^{*}+4\mathrm{diag}\left(Y^{*}\right)=e.$$Moreover,
(11)$$\begin{array}{cc} 1+z_{i}+z_{j}+Z_{ij} & =1+\left(2y_{i}^{*}-1\right)+\left(2y_{j}^{*}-1\right)+1-2y_{i}^{*}-2y_{j}^{*}+4Y_{ij}^{*}\\ & =4Y_{ij}^{*}\geq 0,\;\forall 1\leq i\leq j\leq n. \end{array}$$From Equation (Equation 9) and $e^{T}y^{*}=k$, it follows that
(12)$$z^{T}e={\left(2y^{*}-e\right)}^{T}e=2{\left(y^{*}\right)}^{T}e-n=2k-n.$$Again from Equation (Equation 9) and $e^{T}Y^{*}e=k^{2}$, it is true that
(13)$$\begin{array}{cc} e^{T}Ze & =e^{T}\left(ee^{T}-2e{\left(y^{*}\right)}^{T}-2y^{*}e^{T}+4Y^{*}\right)e\\ & =n^{2}-4kn+4k^{2}={\left(2k-n\right)}^{2}. \end{array}$$From Equation (Equation 9) and Theorem 3 (ii), it holds that
(14)$$\begin{array}{cc} Z-zz^{T} & =ee^{T}-2e{\left(y^{*}\right)}^{T}-2y^{*}e^{T}+4Y^{*}-\left(2y^{*}-e\right){\left(2y^{*}-e\right)}^{T}\\ & =4(Y^{*}-y^{*}{\left(y^{*}\right)}^{T})⪰0. \end{array}$$By Equations (Equation 11)–(Equation 14), we can conclude that (z,Z) defined by Equation (Equation 9) is a feasible solution of the $\left(\mathrm{II}-{\mathrm{SDR}}_{\mathrm{DkS}}\right)$ problem. Furthermore, we have
$$\begin{array}{cc} \frac{1}{8}A•Z & +\frac{1}{4}z^{T}Ae+\frac{1}{8}e^{T}Ae\\ & =\frac{1}{8}A•\left(ee^{T}-2e{\left(y^{*}\right)}^{T}-2y^{*}e^{T}+4Y^{*}\right)+\frac{1}{4}{\left(2y^{*}-e\right)}^{T}Ae+\frac{1}{8}e^{T}Ae\\ & =\frac{1}{2}A•Y^{*}=\mathrm{Opt}({\mathrm{DNNP}}_{\mathrm{DkS}}), \end{array}$$i.e., $\mathrm{Opt}\left({\mathrm{DNNP}}_{\mathrm{DkS}}\right)\leq \mathrm{Opt}\left(\mathrm{II}-{\mathrm{SDR}}_{\mathrm{DkS}}\right)$.Summarizing the analysis above, we obtain $\mathrm{Opt}\left({\mathrm{DNNP}}_{\mathrm{DkS}}\right)=\mathrm{Opt}\left(\mathrm{II}-{\mathrm{SDR}}_{\mathrm{DkS}}\right)$. From Equations (Equation 2) and (Equation 9), we observe that (y,Y) defined by Equation (Equation 2) is an optimal solution for the $\left({\mathrm{DNNP}}_{\mathrm{DkS}}\right)$ problem and (z,Z) defined by Equation (Equation 9) is also an optimal solution for the $\left(\mathrm{II}-{\mathrm{SDR}}_{\mathrm{DkS}}\right)$ problem, respectively. According to Definition 2, we conclude that the $\left({\mathrm{DNNP}}_{\mathrm{DkS}}\right)$ problem and the $\left(\mathrm{II}-{\mathrm{SDR}}_{\mathrm{DkS}}\right)$ problem are equivalent. ☐

The above Theorem 4 shows that $\mathrm{Opt}\left({\mathrm{DNNP}}_{\mathrm{DkS}}\right)=\mathrm{Opt}\left(\mathrm{II}-{\mathrm{SDR}}_{\mathrm{DkS}}\right)$. Note that the $\left({\mathrm{DNNP}}_{\mathrm{DkS}}\right)$ problem has *n* inequality constraints more than the $\left(\mathrm{II}-{\mathrm{SDR}}_{\mathrm{DkS}}\right)$ problem, thus the computational cost of solving $\left({\mathrm{DNNP}}_{\mathrm{DkS}}\right)$ problem may be greater than that of the $\left(\mathrm{II}-{\mathrm{SDR}}_{\mathrm{DkS}}\right)$ problem.

## 4. The Approximation Accuracy

The above section shows that the $\left({\mathrm{DNNP}}_{\mathrm{DkS}}\right)$ problem is equivalent to the $\left(\mathrm{II}-{\mathrm{SDR}}_{\mathrm{DkS}}\right)$ problem which has the tighter upper bound compared to the $\left({\mathrm{SDR}}_{\hat{\mathrm{DkS}}}\right)$ problem (see Theorem 4). In this section, we further investigate the approximation accuracy of the $\left({\mathrm{DNNP}}_{\mathrm{DkS}}\right)$ problem for solving the (DkS) problem, comparing with the standard semidefinite relaxation problems which was proposed in the above sections, under some conditions.

To simplify the expression, we denote $\left[\begin{array}{cc} \frac{1}{8}e^{T}Ae & \frac{1}{8}e^{T}A\\ \frac{1}{8}Ae & \frac{1}{8}A \end{array}\right]$ by *N*. First of all, note that, if $k=\frac{n}{2}$, then the $\left({\mathrm{SDR}}_{\hat{\mathrm{DkS}}}\right)$ problem is simplified to the following problem:$$\tilde{\left({\mathrm{SDR}}_{\hat{\mathrm{DkS}}}\right)}\;\;\;\;\begin{array}{cc} \max & N•\left[\begin{array}{cc} 1 & z^{T}\\ z & Z \end{array}\right],\\ s.t. & \mathrm{diag}(Z)=e,\\ & \left[\begin{array}{cc} 1 & z^{T}\\ z & Z \end{array}\right]\in S_{1+n}^{+}, \end{array}$$and the $\left(\mathrm{II}-{\mathrm{SDR}}_{\mathrm{DkS}}\right)$ problem can be simplified as follows:$$\tilde{\left(\mathrm{II}-{\mathrm{SDR}}_{\mathrm{DkS}}\right)}\;\;\;\;\begin{array}{cc} \max & N•\left[\begin{array}{cc} 1 & z^{T}\\ z & Z \end{array}\right],\\ s.t. & \mathrm{diag}(Z)=e,\\ & 1+z_{i}+z_{j}+Z_{ij}\geq 0,\;\forall 1\leq i\leq j\leq n,\\ & \left[\begin{array}{cc} 1 & z^{T}\\ z & Z \end{array}\right]\in S_{1+n}^{+}. \end{array}$$

Combining Theorem 3 in [25] with the corresponding known approximation accuracy of semidefinite relaxation for some quadratic programming problems [26], we immediately have that the following theorem holds.

**Theorem** **5.**
*If $k=\frac{n}{2}$, then we have*
$$\frac{2}{\pi }\mathrm{Opt}({\mathrm{SDR}}_{\hat{\mathrm{DkS}}})=\frac{2}{\pi }\mathrm{Opt}(\tilde{{\mathrm{SDR}}_{\hat{\mathrm{DkS}}})}\leq \mathrm{Opt}(\mathrm{DkS})=\mathrm{Opt}(\hat{\mathrm{DkS}})\leq \mathrm{Opt}(\tilde{{\mathrm{SDR}}_{\hat{\mathrm{DkS}}})}=\mathrm{Opt}({\mathrm{SDR}}_{\hat{\mathrm{DkS}}})$$
*and*
$$\frac{2}{\pi }\mathrm{Opt}({\mathrm{SDR}}_{\hat{\mathrm{DkS}}})\leq \mathrm{Opt}({\mathrm{DNNP}}_{\mathrm{DkS}})=\mathrm{Opt}\left(\mathrm{II}-{\mathrm{SDR}}_{\mathrm{DkS}}\right)=\mathrm{Opt}\tilde{\left(\mathrm{II}-{\mathrm{SDR}}_{\mathrm{DkS}}\right)}\leq \mathrm{Opt}({\mathrm{SDR}}_{\hat{\mathrm{DkS}}})$$
*for N⪰0. Moreover, if N⪰0 and $N_{ij}\leq 0,\;\forall i\neq j$, we further have*
$$0.87856\mathrm{Opt}({\mathrm{SDR}}_{\hat{\mathrm{DkS}}})\leq \mathrm{Opt}(\mathrm{DkS})=\mathrm{Opt}(\hat{\mathrm{DkS}})\leq \mathrm{Opt}({\mathrm{SDR}}_{\hat{\mathrm{DkS}}})$$
*and*
$$0.87856\mathrm{Opt}({\mathrm{SDR}}_{\hat{\mathrm{DkS}}})\leq \mathrm{Opt}({\mathrm{DNNP}}_{\mathrm{DkS}})=\mathrm{Opt}\left(\mathrm{II}-{\mathrm{SDR}}_{\mathrm{DkS}}\right)\leq \mathrm{Opt}({\mathrm{SDR}}_{\hat{\mathrm{DkS}}})$$
*hold.*


In the following analysis, we assume that $k\neq \frac{n}{2}$. We first observe that
$$\mathrm{Opt}(\mathrm{DkS})=\mathrm{Opt}(\bar{\mathrm{DkS}}),$$since the $\left(\bar{\mathrm{DkS}}\right)$ problem is the homogeneous problem for the (DkS) problem. In addition, note that the constraint $\left[\begin{array}{c} t\\ z \end{array}\right]\in {\left\{-1,\;1\right\}}^{1+n}$ in the $\left(\bar{\mathrm{DkS}}\right)$ problem can be relaxed to the quadratic form $\left[\begin{array}{cc} t & z^{T} \end{array}\right]\left[\begin{array}{c} t\\ z \end{array}\right]=n+1$. Thus, the $\left(\bar{\mathrm{DkS}}\right)$ problem can be relaxed to
$$\tilde{\left(\mathrm{DkS}\right)}\;\;\;\;\begin{array}{cc} \max & \left[\begin{array}{cc} t & z^{T} \end{array}\right]\left[\begin{array}{cc} \frac{1}{8}e^{T}Ae & \frac{1}{8}e^{T}A\\ \frac{1}{8}Ae & \frac{1}{8}A \end{array}\right]\left[\begin{array}{c} t\\ z \end{array}\right],\\ s.t. & \left[\begin{array}{cc} t & z^{T} \end{array}\right]\left[\begin{array}{cc} 0 & \frac{1}{2}e^{T}\\ \frac{1}{2}e & 0 \end{array}\right]\left[\begin{array}{c} t\\ z \end{array}\right]=2k-n,\\ & \left[\begin{array}{cc} t & z^{T} \end{array}\right]\left[\begin{array}{c} t\\ z \end{array}\right]=n+1, \end{array}$$which can further be relaxed to the standard semidefinite programming problem
$$\left({\mathrm{SDR}}_{\tilde{\mathrm{DkS}}}\right)\;\;\;\;\begin{array}{cc} \max & \left[\begin{array}{cc} \frac{1}{8}e^{T}Ae & \frac{1}{8}e^{T}A\\ \frac{1}{8}Ae & \frac{1}{8}A \end{array}\right]•S,\\ s.t. & \left[\begin{array}{cc} 0 & \frac{1}{2}e^{T}\\ \frac{1}{2}e & 0 \end{array}\right]•S=2k-n,\\ & I•S=n+1,\\ & S\in S_{1+n}^{+}. \end{array}$$

Obviously, diag(S)=e implies that $\sum_{i}S_{ii}=n+1$, i.e., I•S=n+1, but we could not obtain diag(S)=e from I•S=n+1. These results further imply that
$$\mathrm{Opt}({\mathrm{SDR}}_{\bar{\mathrm{DkS}}})\leq \mathrm{Opt}({\mathrm{SDR}}_{\tilde{\mathrm{DkS}}}).$$

Similar to the Theorem 4.2 in [27], we have that the following approximation accuracy theorem holds.

**Theorem** **6.**
*$\mathrm{Opt}(\mathrm{DkS})\leq \mathrm{Opt}\tilde{\left(\mathrm{DkS}\right)}\leq \mathrm{Opt}({\mathrm{SDR}}_{\tilde{\mathrm{DkS}}})\leq 2\log\left(67\right)\mathrm{Opt}\tilde{\left(\mathrm{DkS}\right)}$.*


Up to now, we not only establish the equivalence between the $\left({\mathrm{DNNP}}_{\mathrm{DkS}}\right)$ problem and the $\left(\mathrm{II}-{\mathrm{SDR}}_{\mathrm{DkS}}\right)$ problem, but also some approximation accuracy results about the $\left({\mathrm{DNNP}}_{\mathrm{DkS}}\right)$ problem and some standard semidefinite relaxation problems are given. In the following Section Section 5, we will implement some numerical experiments to give a flavour of the actual behaviour of the $\left({\mathrm{DNNP}}_{\mathrm{DkS}}\right)$ problem and some semidefinite relaxation problems.

## 5. Numerical Experiments

In this section, some random (DkS) examples are tested to show the efficiency of the proposed relaxation problems. These relaxation problems are all solved by CVX [28], which is implemented by using MATLAB R2010a on the Windows XP platform, and on a PC with 2.53 GHz CPU. The corresponding comparative numerical results are reported in the following parts.

To give a flavour of the behaviour of the above relaxation problems, we consider results for the following test examples. The data of the test examples are given in Table 1.

The first column of Table 1 denotes the name of the test examples, *n* and *k* stand for the number of vertices of the given graph and the finding subgraph, respectively. The last column denotes the procedures for generating the coefficient matrices *A* in the (DkS) problem. The more detailed explanations for the procedures are given as follows:

• P25. 50 random examples are generated from the ‘seed = 1,2,…,50’. The corresponding coefficient matrices *A* of order n=25 with integer weights are drawn from {0,1,…,10}.

• P30. This example is generated by the MATLAB function randn from the ‘seed = 2012’. The elements of *A* satisfy the standard normal distribution.

• P40. This example is generated by MATLAB function rand from the ‘seed = 2017’. The elements of *A* satisfy the standard uniform distribution on the interval (0,1).

• P50. We generate 50 examples of order n=50 from the ‘seed = 2001,2002,…,2050’. For each coefficient matrix of these examples, half of the entries are randomly set to zero, and the remaining ones are chosen from {−10,−9,…,9,10}.

• P60. This example is generated by MATLAB function rand from the ‘seed = 2020’. The elements of *A* are drawn from {0,1}.

• P80. This example is generated by MATLAB function rand from the ‘seed = 2023’. The coefficient matrix *A* of order n=80 with integer weights drawn from {0,1,…,10}.

First of all, the performances of the $\left({\mathrm{DNNP}}_{\mathrm{DkS}}\right)$ problem and the $\left(\mathrm{II}-S{\mathrm{DR}}_{\mathrm{DkS}}\right)$ problem as well as the $\left(I-{\mathrm{SDR}}_{\mathrm{DkS}}\right)$ problem, for solving P25 and P50, are compared. We use the performance profiles described in Dolan and Moré’s paper [29]. Our profiles are based on optimal values (i.e., average degree) and the number of iterations of these relaxation problems. The Cumulative Probability denotes the cumulative distribution function for the performance ratio within a factor τ∈R, i.e., is the probability that the solver will win over the rest of the solvers. The corresponding comparative results of performance are shown in Figure 1 and Figure 2.

The comparative results for P25 are shown in Figure 1. It is obvious that the $\left({\mathrm{DNNP}}_{\mathrm{DkS}}\right)$ problem and the $\left(\mathrm{II}-{\mathrm{SDR}}_{\mathrm{DkS}}\right)$ problem have the same performance, which is a bit better than that of the $\left(I-{\mathrm{SDR}}_{\mathrm{DkS}}\right)$ problem from the viewpoint of optimal values. In view of the number of iterations, the performance of the $\left(I-{\mathrm{SDR}}_{\mathrm{DkS}}\right)$ problem is the best, and the performance of the $\left(\mathrm{II}-{\mathrm{SDR}}_{\mathrm{DkS}}\right)$ problem is better than that of the $\left({\mathrm{DNNP}}_{\mathrm{DkS}}\right)$ problem.

The performance of the three relaxation problems for solving P50 is shown in Figure 2. The results show that the performance of the $\left({\mathrm{DNNP}}_{\mathrm{DkS}}\right)$ problem is the same as that of the $\left(\mathrm{II}-{\mathrm{SDR}}_{\mathrm{DkS}}\right)$ problem; they are both much better than that of the $\left(I-{\mathrm{SDR}}_{\mathrm{DkS}}\right)$ problem in view of optimal values—although the performance of the $\left(I-{\mathrm{SDR}}_{\mathrm{DkS}}\right)$ problem is better than the one of the $\left({\mathrm{DNNP}}_{\mathrm{DkS}}\right)$ problem and the $\left(\mathrm{II}-{\mathrm{SDR}}_{\mathrm{DkS}}\right)$ problem from the viewpoint of the number of iterations.

All of results show in Figure 1 and Figure 2 further imply that the $\left({\mathrm{DNNP}}_{\mathrm{DkS}}\right)$ problem and the $\left(\mathrm{II}-{\mathrm{SDR}}_{\mathrm{DkS}}\right)$ problem can generate more promising bounds for solving P25 and P50, compared with the $\left(I-{\mathrm{SDR}}_{\mathrm{DkS}}\right)$ problem, while the number of iterations is a bit more. Moreover, the $\left({\mathrm{DNNP}}_{\mathrm{DkS}}\right)$ problem and the $\left(\mathrm{II}-{\mathrm{SDR}}_{\mathrm{DkS}}\right)$ problem have the same performance based on optimal values, although the performance of the $\left(\mathrm{II}-{\mathrm{SDR}}_{\mathrm{DkS}}\right)$ problem is better than that of the $\left({\mathrm{DNNP}}_{\mathrm{DkS}}\right)$ problem from the viewpoint of the number of iterations, for solving P25 and P50.

In order to further show the computational efficiency of the $\left({\mathrm{DNNP}}_{\mathrm{DkS}}\right)$ problem, which is compared with the $\left(\mathrm{II}-{\mathrm{SDR}}_{\mathrm{DkS}}\right)$ problem and some other types of semidefinite relaxation problems proposed in [30], for solving some (DkS) problems. The test examples A50 and A100 are chosen from [30]. (R-20), (R-24) and (R-MET) denote the three semidefinite relaxation problems proposed in [30], respectively. The corresponding numerical results are shown in Table 2, where “−” means that the corresponding information about the number of iterations is not given in [30]. The results show that the computational efficiency of the $\left({\mathrm{DNNP}}_{\mathrm{DkS}}\right)$ problem is better than the one of the $\left(\mathrm{II}-{\mathrm{SDR}}_{\mathrm{DkS}}\right)$ problem from the viewpoints of optimal values and number of iterations, respectively. Note that the performance of the $\left({\mathrm{DNNP}}_{\mathrm{DkS}}\right)$ problem and the $\left(\mathrm{II}-{\mathrm{SDR}}_{\mathrm{DkS}}\right)$ problem are both much better than that of (R−20) and (R−24). Moreover, the performance of the $\left({\mathrm{DNNP}}_{\mathrm{DkS}}\right)$ problem is more competitive with (R−MET) for solving these two problems.

Finally, we further compare the efficiency of the $\left({\mathrm{DNNP}}_{\mathrm{DkS}}\right)$, $\left(\mathrm{II}-{\mathrm{SDR}}_{\mathrm{DkS}}\right)$, $\left(I-{\mathrm{SDR}}_{\mathrm{DkS}}\right)$ and $\left({\mathrm{SDR}}_{\hat{\mathrm{DkS}}}\right)$ problems, for solving examples P30, P40, P60 and P80. The comparative results are shown in Table 3. The results signify that the efficiency of the $\left({\mathrm{DNNP}}_{\mathrm{DkS}}\right)$ problem is always better than that of the $\left(\mathrm{II}-{\mathrm{SDR}}_{\mathrm{DkS}}\right)$ problem from the viewpoint of optimal values and the number of iterations as well as CPU time, respectively, for solving these examples. The performance of the $\left(I-{\mathrm{SDR}}_{\mathrm{DkS}}\right)$ problem and the $\left({\mathrm{SDR}}_{\hat{\mathrm{DkS}}}\right)$ problem are almost the same for solving these examples. Moreover, note that the optimal value of the $\left({\mathrm{DNNP}}_{\mathrm{DkS}}\right)$ problem for solving P80 is larger than that of the $\left(\mathrm{II}-{\mathrm{SDR}}_{\mathrm{DkS}}\right)$ problem. Thus, we can conclude that it may be more promising to use the $\left({\mathrm{DNNP}}_{\mathrm{DkS}}\right)$ problem than to use the $\left(\mathrm{II}-{\mathrm{SDR}}_{\mathrm{DkS}}\right)$ problem for solving some specific (DkS) problems in practice.

## 6. Conclusions

In this paper, the DkS problem is studied, whose goal is to find a *k*-vertex subgraph such that the total weight of edges in this subgraph is maximized. This problem is NP-hard on bipartite graphs, chordal graphs, and planar graphs. By using the advantages of the structure of the DkS problem, the doubly nonnegative relaxation and the new semidefinite relaxation with tighter relaxation for solving the DkS problem are established, respectively. Moreover, we prove that the two relaxation problems are equivalent under the suitable conditions, and give some approximation accuracy results for these relaxation problems. Finally, the comparative numerical results show that the efficiency of the doubly nonnegative relaxation is better than the one of semidefinite relaxation for solving some DkS problems.

## Figures and Tables

**Figure 1 entropy-21-00108-f001:**
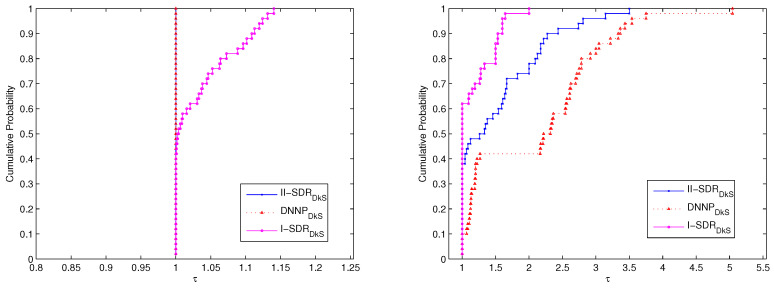
The performance of the $\left({\mathrm{DNNP}}_{\mathrm{DkS}}\right)$ problem and the $\left(\mathrm{II}-{\mathrm{SDR}}_{\mathrm{DkS}}\right)$ problem as well as the $\left(I-{\mathrm{SDR}}_{\mathrm{DkS}}\right)$ problem for solving P25. The performance profile of (**left**) figure is based on optimal values, and (**right**) figure is based on the number of iterations.

**Figure 2 entropy-21-00108-f002:**
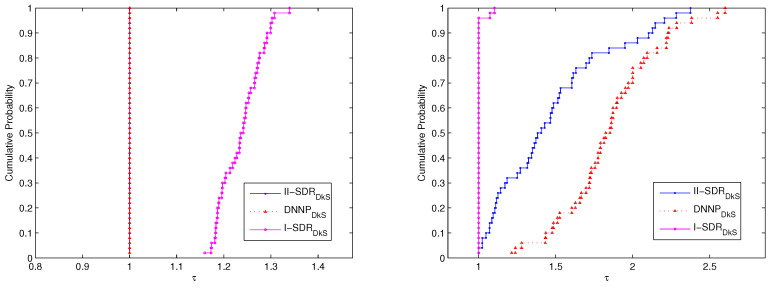
The performance of the $\left({\mathrm{DNNP}}_{\mathrm{DkS}}\right)$ problem and the $\left(\mathrm{II}-{\mathrm{SDR}}_{\mathrm{DkS}}\right)$ problem as well as $\left(I-{\mathrm{SDR}}_{\mathrm{DkS}}\right)$ problem for solving P50. The performance profile in the (**left**) figure is according to optimal values, and the (**right**) figure is based on the number of iterations.

**Table 1 entropy-21-00108-t001:** Data of the test examples.

P25	*n* = 25;	*k* = 5;	seed = 1, 2, …, 50; rand(‘seed’,seed);
			A = round(10 × rand(*n*)); A = tril(A,−1) + triu(A’,0);
P30	n= 30;	k= 10;	seed = 2012; randn(‘seed’,seed); A = randn(n,n);
			A = tril(A,−1) + triu(A’,0);
P40	n= 40;	k= 15;	seed = 2017; rand(‘seed’,seed); A = rand(n,n);
			A = tril(A,−1) + triu(A’,0);
P50	n= 50;	k= 10;	seed = 2001, 2002, …, 2050; rand(‘seed’,seed);
			G = (rand(*n*)<0.5); A = round(20 × rand(*n*)−10); A = triu(A,1);
			A = A.*G; A100 = A + A’;
P60	n= 60;	k= 20;	seed = 2020; rand(‘seed’,seed); A = round(rand(n,n));
			A = tril(A,−1) + triu(A’,0);
P80	n= 80;	k= 30;	seed = 2023; rand(‘seed’,seed); A = round(rand(n,n));
			A = tril(A,−1) + triu(A’,0);

**Table 2 entropy-21-00108-t002:** Numerical results for some DkS examples in [30].

Example	Solving Model	Optimal Value	Number of Iterations
A50	$\left({\mathrm{DNNP}}_{\mathrm{DkS}}\right)$	337.4	51
	$\left(\mathrm{II}-{\mathrm{SDR}}_{\mathrm{DkS}}\right)$	337.416	57
	(R−20)	365.508	−
	(R−24)	365.831	−
	(R−MET)	333.65	−
A100	$\left({\mathrm{DNNP}}_{\mathrm{DkS}}\right)$	505.989	54
	$\left(\mathrm{II}-{\mathrm{SDR}}_{\mathrm{DkS}}\right)$	506.126	68
	(R−20)	607.691	−
	(R−24)	611.070	−
	(R−MET)	490.50	−

**Table 3 entropy-21-00108-t003:** Numerical results for some (DkS) examples.

Example	Solving Model	Optimal Value	Number of Iterations	CPU Time (seconds)
P30	$\left({\mathrm{DNNP}}_{\mathrm{DkS}}\right)$	30.5894	52	3.4
	$\left(\mathrm{II}-{\mathrm{SDR}}_{\mathrm{DkS}}\right)$	30.5903	70	5.0
	$\left(I-{\mathrm{SDR}}_{\mathrm{DkS}}\right)$	33.3382	33	0.7
	$\left({\mathrm{SDR}}_{\hat{\mathrm{DkS}}}\right)$	33.3383	34	0.8
P40	$\left({\mathrm{DNNP}}_{\mathrm{DkS}}\right)$	74.4657	50	14.7
	$\left(\mathrm{II}-{\mathrm{SDR}}_{\mathrm{DkS}}\right)$	74.4658	66	20.6
	$\left(I-{\mathrm{SDR}}_{\mathrm{DkS}}\right)$	75.5198	33	1.5
	$\left({\mathrm{SDR}}_{\hat{\mathrm{DkS}}}\right)$	75.5197	34	1.6
P60	$\left({\mathrm{DNNP}}_{\mathrm{DkS}}\right)$	150.385	52	105.5
	$\left(\mathrm{II}-{\mathrm{SDR}}_{\mathrm{DkS}}\right)$	150.387	60	125.3
	$\left(I-{\mathrm{SDR}}_{\mathrm{DkS}}\right)$	154.272	40	3.2
	$\left({\mathrm{SDR}}_{\hat{\mathrm{DkS}}}\right)$	154.273	35	2.2
P80	$\left({\mathrm{DNNP}}_{\mathrm{DkS}}\right)$	230.912	84	739.9
	$\left(\mathrm{II}-{\mathrm{SDR}}_{\mathrm{DkS}}\right)$	312.644	85	770.4
	$\left(I-{\mathrm{SDR}}_{\mathrm{DkS}}\right)$	318.945	41	5.7
	$\left({\mathrm{SDR}}_{\hat{\mathrm{DkS}}}\right)$	318.945	34	3.2

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
