# Peer review of "Doubly Nonnegative and Semidefinite Relaxations for the Densest k-Subgraph Problem"

_entropy, 2019, doi:10.3390/e21020108_

Round 1
Reviewer 1 Report
The paper suggests two relaxations for a central combinatorial optimization problem - densest k-subgraph. The paper presents the two relaxations, discusses some properties, and then provides some numerical testing.
some concrete remarks:
1. The write-up, grammar, wording, phrasing must be improved. In many places "the" "a" is missing, sentences are misworded, etc.
2. In general, avoid saying "tighter bound" unless you prove it. All you prove is a weak inequality, and it could be that both relaxations are equal on all instances. You can use "tighter relaxation" meaning that you impose more constraints, but you cannot say anything about the value of the relaxation.
3. Page 2 line 50 - Sn++ is the cone of positive *definite* matrices ("definite" is missing)
4. Page 2 line 62: Even if A was PSD matrix the problem is still NPH because the search space is not convex (y is discrete).
5. Page 3 line 65-66: what is K in the definition of C1+n
6. Page 4 line 90: (ii): what you write doesn't make sense ... you optimize over the smaller set - S1+n \cap N1+n, and you say that since it is contained in a convex set, then CVX can solve it ...
in the same way, you can say that the discrete set y={0,1}^n is contained in y\in R^n, so you can optimize over it.
7. Page 4 line 101: again, avoid saying that the bound is tighter... say that it is not larger ...
8. Section 4 is very weak ... if the discussion is valid only for k=n/2 then it is useless and not interesting. Either improve the proof or remove the section.
9. The numerical test part is hard to follow. Please say clearly what is the average degree of the entire graph and what is the average degree of the planted dense subgraph (at least in expectation).
Also, please compare to a simple greedy method: for example, as long as you have more than k vertices in the graph, throw away the vertex with the lowest degree. Now show how your relaxations compare with this simple greedy heuristic.
Also, the fact that you can only solve graphs with n=80 vertices, and then CVX runs forever, means that your approach is totally not applicable to real-life scenario ...
Author Response
Response to Reviewer 1 Comments
Dear Reviewer 1,
Thank you very much for your valuable comments and suggestions on the original version of our manuscript. We have taken your comments and suggestions into careful consideration in the revision of our manuscript. In particular, the presentation has been substantially revised and improved. Our responses to the comments and suggestions are detailed below.
We look forward to hearing from you soon and give our best wishes to you.
Yours faithfully,
Chuan-Hao Guo, Yuan Guo and Bei-Bei Liu
Point 1: The write-up, grammar, wording, phrasing must be improved. In many places "the" "a" is missing, sentences are misworded, etc.
Response 1: Thank you for your valuable comments. We have checked the whole manuscript very carefully, and corrected some grammar, wording mistakes accordingly. We also do our best to polish it with the help of the two colleagues who are completely fluent in English.
Point 2: In general, avoid saying "tighter bound" unless you prove it. All you prove is a weak inequality, and it could be that both relaxations are equal on all instances. You can use "tighter relaxation" meaning that you impose more constraints, but you cannot say anything about the value of the relaxation.
Response 2: We would like to express our appreciation to you for suggesting how to improve our manuscript. We have modified and polished the related sentences in the revised manuscript.
Point 3: Page 2 line 50 - Sn++ is the cone of positive definite matrices ("definite" is missing).
Response 3: We have added “definite” in the revised manuscript.
Point 4: Page 2 line 62: Even if A was PSD matrix the problem is still NPH because the search space is not convex (y is discrete).
Response 4: Thank you for your suggestions. We have corrected this mistake in the revised manuscript.
Point 5: Page 3 line 65-66: what is K in the definition of C1+n.
Response 5: Thank you very much for pointing out this mistake. k/K has been replaced by another letter h/H in the definition of C1+n.
Point 6: Page 4 line 90: (ii): what you write doesn't make sense ... you optimize over the smaller set - S1+n \cap N1+n, and you say that since it is contained in a convex set, then CVX can solve it ... in the same way, you can say that the discrete set y={0,1}^n is contained in y\in R^n, so you can optimize over it.
Response 6: Thank you very much for your valuable comments. First of all, we have rewritten the related improper expression in the Remark 2.1. Secondly, it is well-known that CVX is a modeling system for convex optimization which including semidefinite programs. Since S+1+n∩N+1+n is the special subset of S+1+n, meanwhile, the types of variable contained in the both sets are the same, it means that (DNNPDkS) problem could also be solved by some softwares for semidefinite programs, such as CVX. Therefore we can say CVX could be used to solve (DNNPDkS) problem. The type of variables in {0,1}n is discrete, however, the variables type in Rn is continuous, they are completely different types of variables. In general, we cannot use the method for optimize over Rn directly to optimize over {0,1}n. Finally, we would like to express our appreciation to you once again for suggesting how to improve our manuscript.
Point 7: Page 4 line 101: again, avoid saying that the bound is tighter... say that it is not larger ...
Response 7: This mistake has been corrected in the revised manuscript.
Point 8: Section 4 is very weak ... if the discussion is valid only for k=n/2 then it is useless and not interesting. Either improve the proof or remove the section.
Response 8: Thank you very much for your valuable suggestions. Yes, the presented approximation accuracy result of (DNNPDkS) problem, is the same as the one of (II-SDRDkS) problem for k=n/2 with the best known results, could be obtained by simple calculation according to the results in the references [26, 27]. Since the process of proof is elementary, thus it is omitted in the manuscript. When k≠n/2, there are not any approximation results for (DNNP) problem yet. It also gives us an opportunity or direction to go on with studying in this field.
Point 9: The numerical test part is hard to follow. Please say clearly what is the average degree of the entire graph and what is the average degree of the planted dense subgraph (at least in expectation).
Also, please compare to a simple greedy method: for example, as long as you have more than k vertices in the graph, throw away the vertex with the lowest degree. Now show how your relaxations compare with this simple greedy heuristic.
Also, the fact that you can only solve graphs with n=80 vertices, and then CVX runs forever, means that your approach is totally not applicable to real-life scenario ...
Response 9: Thanks for your consideration for the numerical tests. First of all, some improper or unclear expressions are rewritten in the revised manuscript. Secondly, our current manuscript mainly focus on to show the potential promising of (DNNPDkS) problem, are not to design new methods, for solving some DkS problems. Of course, the performance of (DNNPDkS) problem is also compared with the one of (R-20) and (R-24) as well as (R-MET), which are proposed in the reference [30], for solving the given two DkS problems. The comparative results also show that the performance of (DNNPDkS) problem is more promising and competitive. Besides, in our another manuscript about how to find/solve a densest k-subhypergraph, by using the technique of doubly nonnegative relaxation, for the given hypergraph, we have designed a new method to solve the doubly nonnegative relaxation problem for densest k-subhypergraph. Some comparative results between our method with some greedy heuristics for solving some densest k-subhypergraph will also be presented in it. The elementary results are interesting, and we will submit it to the journal when it is finished.
In the numerical experiments, we have only tested the DkS problem with up to 80 vertices by CVX. This does not mean that CXV will run forever when the number of vertices for DkS problem is more than 80. For example, for DkS problems with 300 vertices, the results will be returned in a certain amount of time. Moreover, the comparative results of (DNNPDkS) and (II-SDRDkS) in Table 2 are also show that CVX can efficiently solve problem A100 with 100 vertices. At the same time, we are going to establish the related doubly nonnegative programming model, for some densest k-subhypergraph which be chosen from some social networks, in our another manuscript.
Finally, we would like to express our great appreciation to you for your comments on our manuscript, and we also desire you will not reject our manuscript for this comment.

Reviewer 2 Report
This manuscript presented “Doubly Nonnegative and Semidefinite Relaxations
for Densest k-Subgraph Problem”. Although it is very interesting but revision is required.
1. Why are you using Doubly nonnegative relaxation method? Mentioned the significance of this method with latest references.
2. What are the application of this present model work. Please explained it.
3. In the introduction section the references are not in sequence format. Such as [1], [2] and so on…
4. Nomenclature must be added in the manuscript.
5. Grammatical mistakes must be corrected by taking help from any English native speaker.
Author Response
Response to Reviewer 2 Comments
Dear Reviewer 2,
Thank you very much for your valuable comments and suggestions on the original version of our manuscript. We have taken your comments and suggestions into careful consideration in the revision of our manuscript. In particular, the presentation has been substantially revised and improved. Our responses to the comments and suggestions are detailed below.
We look forward to hearing from you soon and give our best wishes to you.
Yours faithfully,
Chuan-Hao Guo, Yuan Guo and Bei-Bei Liu
Point 1: Why are you using Doubly nonnegative relaxation method? Mentioned the significance of this method with latest references.
Response 1: Thank you very much for your valuable comments. First, we have shown that the doubly nonnegative relaxation (DNNPDkS) and the semidefinite relaxation (II-SDRDkS) are equivalent (Theorem 3.2) under some conditions, although the numerical results have also shown that the doubly nonnegative relaxation (DNNPDkS) is more promising and effective for solving some DkS problems. Moreover, the performance of (DNNPDkS) is more competitive with (R-20) and (R-24) as well as (R-MET), which are proposed in the reference [30], for solving the given two DkS problems. We have also added some references [14,16,18] about the applications of optimization problems over the doubly nonnegative cone.
Finally, we would like to express our appreciation to you for suggesting how to improve our manuscript.
Point 2: What are the application of this present model work. Please explained it.
Response 2: Thank you very much for your valuable comments. First of all, we have added some brief introductions of optimization problems over the doubly nonnegative cone, for example, as a strengthening of the Lovasz-υ-number for approximating the largest clique in a graph [Math. Program. Ser. A 2014], solving multiple objective quadratic programming problems [J. Ind. Manag. Optim. 2014] and some other applications one may refer to the references [15,17,18,19] in the section of Introduction. Moreover, the numerical results in our current paper have also shown that the doubly nonnegative relaxation is more promising than the semidefinite relaxation for solving some DkS problems.
Due to the vast application of DkS problem, such as facility location, community detection in social networks as well as identifying protein families and molecular complexes in protein-protein interaction networks (which are usually solved by some greedy heuristics), we are now working on an manuscript about how to find/solve a densest k-subhypergraph, by using the technique of doubly nonnegative relaxation, for the given hypergraph which is widely applied in some social networks. The elementary results are interesting, and we will submit it to the journal when it is finished.
Point 3: In the introduction section the references are not in sequence format. Such as [1], [2] and so on …
Response 3: Thank you for your comments. We have rearranged the references, according to the sequence format, in the revised manuscript.
Point 4: Nomenclature must be added in the manuscript.
Response 4: Thank you for your valuable comments. We have added the corresponding nomenclature in the revised manuscript in a table in Page 2.
Point 5: Grammatical mistakes must be corrected by taking help from any English native speaker.
Response 5: We greatly appreciate both your comments and that of the referee concerning improvement to our paper. We have checked the whole manuscript very carefully, and corrected some grammatical mistakes accordingly. We also do our best to polish it with the help of the two colleagues who are completely fluent in English. We hope that the revised manuscript is now suitable for publication in the journal.
Once again, thank you very much for your valuable comments and suggestions.

Round 2
Reviewer 2 Report
The revised paper could be accepted in present form.